# Prognostic Value of Computed Tomography-Derived Muscle Density for Postoperative Complications in Enhanced Recovery After Surgery (ERAS) and Non-ERAS Patients

**DOI:** 10.3390/nu17142264

**Published:** 2025-07-09

**Authors:** Fiorella X. Palmas, Marta Ricart, Amador Lluch, Fernanda Mucarzel, Raul Cartiel, Alba Zabalegui, Elena Barrera, Nuria Roson, Aitor Rodriguez, Eloy Espin-Basany, Rosa M. Burgos

**Affiliations:** 1Nutrition Support Unit, Endocrinology and Nutrition Department, University Hospital Vall d’Hebron, 08035 Barcelona, Spain; marta.ricart@vallhebron.cat (M.R.); amador.lluch@vallhebron.cat (A.L.); fernanda.mucarzel@vhir.org (F.M.); raul.cartiel@vhir.org (R.C.); alba.zabalegui@vallhebron.cat (A.Z.); rosa.burgos@vallhebron.cat (R.M.B.); 2Diabetes and Metabolism Research Unit, Vall d’Hebron Institut de Recerca (VHIR), 08035 Barcelona, Spain; 3Department of Medicine, Universitat Autònoma de Barcelona, 08193 Barcelona, Spain; 4ARTIS Development, 35017 Las Palmas de Gran Canaria, Spain; ebarrera@artisdevelopment.com; 5Department of Radiology, Institut De Diagnòstic Per La Imatge (IDI), University Hospital Vall d’Hebron, 08035 Barcelona, Spain; nuria.roson.idi@gencat.cat (N.R.); aitor.rodriguez@vhir.org (A.R.); 6Unit of Colorectal Surgery, Department of General and Digestive Surgery, University Hospital Vall d’Hebron, Universitat Autònoma de Barcelona, 08035 Barcelona, Spain; eloy.espin@vallhebron.cat

**Keywords:** surgery, colorectal cancer, ERAS, computed tomography, Hounsfield units, body composition

## Abstract

**Background**: Prehabilitation programs improve postoperative outcomes in vulnerable patients undergoing major surgery. However, current screening tools such as the Malnutrition Universal Screening Tool (MUST) may lack the sensitivity needed to identify those who would benefit most. Muscle quality assessed by Computed Tomography (CT), specifically muscle radiodensity in Hounsfield Units (HUs), has emerged as a promising alternative for risk stratification. **Objective**: To evaluate the prognostic performance of CT-derived muscle radiodensity in predicting adverse postoperative outcomes in colorectal cancer patients, and to compare it with the performance of the MUST score. **Methods**: This single-center cross-sectional study included 201 patients with non-metastatic colon cancer undergoing elective laparoscopic resection. Patients were stratified based on enrollment in a multimodal prehabilitation program, either within an Enhanced Recovery After Surgery (ERAS) protocol or a non-ERAS pathway. Nutritional status was assessed using MUST, SARC-F questionnaire (strength, assistance with walking, rise from a chair, climb stairs, and falls), and the Global Leadership Initiative on Malnutrition (GLIM) criteria. CT scans at the L3 level were analyzed using automated segmentation to extract muscle area and radiodensity. Postoperative complications and hospital stay were compared across nutritional screening tools and CT-derived metrics. **Results**: MUST shows limited sensitivity (<27%) for predicting complications and prolonged hospitalization. In contrast, CT-derived muscle radiodensity demonstrates higher discriminative power (AUC 0.62–0.69), especially using a 37 HU threshold. In the non-ERAS group, patients with HU ≤ 37 had significantly more complications (33% vs. 15%, *p* = 0.036), longer surgeries, and more severe events (Clavien–Dindo ≥ 3). **Conclusions**: Opportunistic CT-based assessment of muscle radiodensity outperforms traditional screening tools in identifying patients at risk of poor postoperative outcomes, and may enhance patient selection for prehabilitation strategies like the ERAS program.

## 1. Introduction

Surgical prehabilitation is an increasingly widespread strategy aimed at optimizing the physical and emotional condition of patients before surgery, reducing the risk of postoperative complications, and improving recovery and long-term outcomes [1,2,3,4]. Given its growing implementation, and the lack of unlimited resources, it is essential to appropriately select the patients who are most likely to benefit from these interventions. In this regard, the Enhanced Recovery After Surgery (ERAS) Society recommends systematic nutritional screening during the preoperative period to identify patients at nutritional risk and refer them for comprehensive nutritional assessment and appropriate medical-nutritional treatment [5].

However, although there is no consensus on the recommended method of screening and nutritional referral, the Enhanced Recovery After Surgery (ERAS) society strongly recommends the use of the Malnutrition Universal Screening Tool (MUST) for screening before surgery [5]. Additionally, it is important to recognize that nutritional screening is not infallible and can lead to false negatives and false positives, highlighting the need for more sensitive and specific assessment methods [6,7].

Most conventional nutritional screening tools, such as the Malnutrition Universal Screening Tool (MUST), do not include an assessment of body composition (BC) or muscle mass, which may lead to false negatives and missed diagnoses of malnutrition or sarcopenia [8,9]. In clinical practice, it is essential to assess BC to detect muscle abnormalities that can exist even in patients with normal body weight or Body Mass Index (BMI) [10,11,12]. In response to this limitation, the Global Leadership Initiative on Malnutrition (GLIM) criteria were proposed and are currently considered the most widely accepted tool for diagnosing malnutrition [13]. Among its phenotypic criteria, muscle mass assessment is explicitly included, making it a more accurate and sensitive tool for nutritional diagnosis [14].

Both conditions—malnutrition and sarcopenia—are closely related to poor clinical outcomes and worse prognosis in oncological patients, especially those undergoing surgery [15,16]. For this reason, ongoing efforts are attempting to unify the diagnostic criteria for sarcopenia, just as was previously done for malnutrition through the GLIM initiative [17,18].

Among the validated techniques for assessing body composition (BC) are Dual-Energy X-ray Absorptiometry (DEXA), Bioelectrical Impedance Analysis (BIA), ultrasound, Magnetic Resonance Imaging (MRI), and Computed Tomography (CT) [19]. In recent years, CT has emerged as a reference technique, not only for quantifying muscle area but also for evaluating muscle quality based on radiodensity in Hounsfield Units (HUs) [20,21]. In oncology patients, CT-derived muscle quality—measured through HU—has consistently shown strong associations with postoperative complications and is considered an independent prognostic factor (Figure 1) [22,23,24].

Among the types of cancer where CT imaging is routinely performed for diagnosis, staging, and follow-up, colorectal cancer (CRC) stands out as a particularly relevant case. As the third most commonly diagnosed cancer globally and the second leading cause of cancer-related death [25], CRC represents a clinical context in which the integration of surgical prehabilitation and body composition assessment through opportunistic CT may substantially improve patient outcomes. In 2020 alone, nearly 2 million new cases of CRC were diagnosed and over 900,000 deaths occurred, with future projections indicating even higher incidence and mortality [25].

Importantly, the prevalence of malnutrition in CRC patients ranges from 7.5% to 45.1% [26], while sarcopenia may affect up to 60% of them, including a 15% prevalence of sarcopenic obesity [27]. Given this, CT is routinely used in the diagnosis, staging, and monitoring of CRC, offering an invaluable opportunity for assessing nutritional status via opportunistic CT analysis [28,29,30]. Despite this potential, the use of CT as a screening tool in nutritional prehabilitation units remains underexplored. This study aims to address this knowledge gap by evaluating the utility and feasibility of CT as a screening and referral tool for nutritional prehabilitation in patients undergoing colorectal cancer surgery.

This study aims to evaluate the role of opportunistic CT as a screening tool in the context of surgical prehabilitation in patients with non-metastatic colorectal cancer undergoing elective surgery. It seeks to characterize the nutritional status and body composition of patients enrolled in a multimodal prehabilitation program compared to those not receiving such intervention. It also assesses the ability of the MUST to identify patients at higher risk of postoperative complications.

We hypothesize that CT-derived muscle radiodensity is a more accurate predictor of adverse postoperative outcomes in colorectal cancer patients than traditional nutritional screening tools such as MUST, and may serve as a more effective criterion for identifying candidates for prehabilitation programs.

## 2. Materials and Methods

The study was conducted in accordance with the Declaration of Helsinki. The research protocol was approved by the Research Ethics Committee for Medicines of Vall d’Hebron University Hospital (reference number PR(AG)489/2021; approval date: 29 October 2021). Written informed consent was obtained from all participants prior to their inclusion in the study.

### 2.1. Patient Selection

We performed a single-centre cross-sectional study including consecutive patients diagnosed with colon cancer who underwent laparoscopic oncological surgery. This study was conducted at a tertiary hospital in Spain, between April 2021 and December 2023. Inclusion criteria: (a) more than 18 years of age; (b) diagnosis of colon cancer confirmed by biopsy; (c) acceptance and return of signed informed consent after clarification of doubts; (d) abdominal CT scan conducted within 60 days before surgical intervention. Exclusion criteria: (a) unable to perform CT scan; (b) patients who did not undergo surgery; (c) patients who underwent open surgery instead of laparoscopic surgery. Two well-differentiated study groups were obtained: patients included in the prehabilitation program before surgery (ERAS) and Non-ERAS group. Patients prehabilitated before surgery were categorized in this group after being included in the ERAS program to receive a multimodal prehabilitation (nutrition, psychology and physiotherapy). Patients can be included in this group in two different ways: (i) patients who are referred directly to prehabilitation by the surgical team in case they are considered to be a highly complex surgery, or patients with a high number of comorbidities or fragile patients; (ii) patients who are referred directly to the anesthesia team and are subsequently screened by the advanced practice nursing team with the MUST test, the Fatigue, Resistance, Ambulation, Illnesses and Loss of weight (FRAIL) test and the Patient Health Questionnaire-4 (PHQ4) test. If the screening is positive (MUST > 2, FRAIL > 0, PHQ4 > 3), patients are referred to multimodal prehabilitation. This clinical decision process is illustrated in Figure 2.

Then Non-ERAS group, including patients from the Tumours Committee, directly referred to the anesthesia team and subsequently screened by the advanced practice nursing team with different tests. If MUST < 2, the FRAIL test = 0 and/or the PHQ4 < 3, the patients are not referred to prehabilitation and undergo surgery directly.

### 2.2. Clinical Data Collection

Patients were recruited 48 h after colon oncology surgery and after signing the informed consent. Screening of malnutrition and sarcopenia was done by using the MUST screening tool and SARC-F questionnaire, respectively. To assess nutritional status, the following variables were recorded: height (m), weight (kg), Body Mass Index (BMI), and percentage of weight loss in the previous six months. Body composition was evaluated using a portable bioelectrical impedance device (Bodystat Ltd., Douglas, Isle of Man, UK), collecting Fat-Free Mass Index (FFMI), phase angle, and Body Cell Mass (BCM).

Appendicular Skeletal Muscle Mass (ASMM) was estimated using the equations proposed by Sergi et al. [31] for patients over 60 years of age and by Kyle et al. [32] for those aged 60 or younger, and was used as an indicator of low muscle mass.

Malnutrition was diagnosed according to the GLIM criteria. All patients were considered to meet at least one etiologic criterion due to the presence of colorectal cancer, a chronic inflammatory disease. Phenotypic criteria included low BMI, weight loss > 5% over six months, and low FFMI by BIA, based on the cutoff values recommended by the GLIM consensus. Oncological variables were documented, including the type of neoplasia and disease stage. The hospital stay was classified as normal (<7 days) or prolonged (>10 days). Stays of 7 to 9 days were excluded, as discharge decisions within this range are often influenced by non-clinical factors. This approach improves the clarity and interpretability of the results by focusing on patient groups with well-defined clinical trajectories [33].

### 2.3. Body Composition Assessment by Skeletal Computed Tomography (CT)

To assess skeletal muscle and abdominal adipose tissue area, we use FocusedON-BC software Version 2.1 to analyze transverse CT images taken at the third lumbar vertebra (L3) using a multidetector CT scanner (Aquilion Prime SP, Canon Medical Systems, Otawara, Japan), with the following technical parameters: 135 kV (tube voltage), 1 mm 80 row (detector configuration). All patients underwent diagnostic CT with intravenous contrast, ensuring they were performed within two months prior to our evaluation. The muscle groups analyzed include the psoas, erector spinae, quadratus lumborum, transversus abdominis, external and internal obliques, and rectus abdominis. In addition, we evaluated adipose tissue and classified it into subcutaneous, visceral, and intermuscular fat. Our recorded variables include Skeletal Muscle Area (SMA) in both cm^2^ and %; Skeletal Muscle Index (SMI) in cm^2^/m^2^; Intermuscular Adipose Tissue (IMAT) area in both cm^2^ and %; Intermuscular Adipose Tissue Index (IIMAT) in cm^2^/m^2^; Visceral Fat Area (VFA) in both cm^2^ and %; Subcutaneous Fat Area (SFA) in both cm^2^ and %; Visceral Fat Index (VFI) in cm^2^/m^2^; Subcutaneous Fat Index (SFI) in cm^2^/m^2^; and the mean Hounsfield Units (HUs) value for each segmented tissue. In addition, tissue quality was assessed based on its average HU value. Standard thresholds were used as follows: −29 to 150 HU for skeletal muscle, −190 to −30 for subcutaneous and intermuscular adipose tissue and −150 to −50 for visceral adipose tissue.

### 2.4. Statistical Analysis

This study evaluates the possibility of using body composition information extracted from CT images as a screening tool in patients with surgical indications to identify those at nutritional risk that would result in a poor clinical outcome. The lack of similar precedents justifies our exploratory approach with a smaller sample size as a pilot study. Statistical analysis was carried out using Python Software Foundation 3.12. Continuous variables are presented as mean ± Standard Deviation (SD) for normally distributed variables and median and Interquartile Range (IQR) for non-normally distributed variables. Categorical variables are presented using percentages. Statistical significance was accepted at *p* < 0.05. Patients at nutritional risk were determined as those with a hospital stay longer than 9 days, as it is associated with poor clinical outcomes. Patients with a hospital stay less than 7 days had a good clinical outcome. It is important to highlight the 7 to 9 days interval, as it cannot be confidently associated with either outcome and may even be influenced by logistical factors. Consequently, patients whose hospital stays fell within 7 to 9 days were excluded from the sample for this reason. The same statistical analysis was also performed using postoperative complications as an indicator of poorer clinical outcomes in the statistical analysis. The usefulness of both MUST and the proposed body composition information to identify patients at nutritional risk that would result in poor clinical outcomes was evaluated based on their sensibility and specificity. The best cut-off points for continuous variables were selected from the ROC curves as those that minimize distance to the (0, 1) point. The Variance Inflation Factor (VIF) was also calculated to verify that the body composition variables extracted from the CT were able to provide new information that was not contained in the variables included in the MUST (such as age, weight and height).

## 3. Results

As shown in Table 1, a total of 201 patients (median age: 73.0 years, IQR: 64.8–81.6) who underwent laparoscopic surgery for colon cancer were recruited. Of these, 71 (35.3%) were included in the Enhanced Recovery After Surgery (ERAS) program, while 130 (64.7%) were not. A significant statistical difference (*p* < 0.05) was observed between both groups; therefore, all statistical analyses were conducted separately for each group. In particular, clinical differences and outcomes may be attenuated in the ERAS group as a result of the program’s own intervention. The sample comprised 120 men (60.0%) and 81 women (40.0%), with no statistically significant differences in the sex distribution between the ERAS and non-ERAS groups. All recruited patients had tumours located in the colon and underwent laparoscopic surgery. The surgery duration was significantly longer in the ERAS group compared to the non-ERAS group (median: 262 min, IQR: 228–307 vs. median: 240 min, IQR: 217.5–286.5; *p* < 0.05). Similarly, patients in the ERAS group had a longer hospital stay than those in the non-ERAS group (median: 6 days, IQR: 4–11 vs. median: 4 days, IQR: 3–5.25; *p* < 0.05). Postoperative complications were more frequent in the ERAS group (49% vs. 21%; *p* < 0.001), and mortality was also higher (14% vs. 5%; *p* = 0.036).

Firstly, the sensitivity and specificity of the MUST score in identifying patients with a higher risk of complications or prolonged hospital stays were evaluated. The MUST score was dichotomized by comparing MUST = 0 versus MUST > 0 for both ERAS and non-ERAS groups. The results are presented in Table 2. As shown, sensitivity was consistently low (<27%) across all scenarios, limiting the usefulness of the MUST score as a screening tool. It is important to note that, in clinical practice, the commonly used threshold for the MUST score is >2; however, raising the threshold further reduces its sensitivity. Following the finding that the MUST score has limited utility as a screening tool to predict complications (low sensibility), we assessed whether body composition data derived from CT images could provide additional, independent information beyond the primary variables used in the MUST criteria (i.e., weight, height and age). To investigate this, we conducted a Variance Inflation Factor (VIF) analysis that included these three primary variables along with two CT-derived parameters: skeletal muscle area (in cm^2^) and average skeletal muscle radiodensity (in Hounsfield Units). To evaluate potential multicollinearity among the independent variables included in the regression model, a VIF analysis was also performed. All VIF values were below the commonly accepted threshold of 3, indicating no collinearity. Specifically, the VIF values were as follows: age (1.25), weight (2.41), height (1.62), muscle quantity (2.60), and muscle average radiodensity in HU (1.67). In particular, derived variables such as Body Mass Index (BMI), Skeletal Muscle Index (SMI), and relative muscle area (in %) were excluded from the analysis, as they represent linear combinations of other included variables (BMI = weight/height^2^; SMI = muscle area/height^2^; muscle % = muscle area/region of interest area). Including such variables would have artificially increased the VIF values and introduced redundancy into the model.

Since CT-derived variables (skeletal muscle area and average skeletal muscle radiodensity) were shown to provide independent information relative to the primary MUST variables (weight, height, and age), we evaluated each variable individually to determine their ability to identify patients at risk of poor outcomes, as defined in our study. Receiver Operating Characteristic (ROC) curve analyses were conducted for each variable, and the Area Under the Curve (AUC) was calculated. The optimal cut-off point for each variable was defined as the point with the smallest Euclidean distance to (0, 1) on the ROC curve. Then, the sensitivity and the specificity were determined. The results are presented in Table 3 and Table 4. According to the obtained results, muscle radiodensity (in HU) demonstrated the highest overall discriminative capacity, with AUC values ranging from 0.620 to 0.692. The lowest sensitivity (57.89%) was observed in the analysis conducted for the non-ERAS group based on hospital stay, with a specificity of 78.85% and an optimal cut-off point of 34.46 HU. In contrast, the highest sensitivity (70.83%) was obtained in the analysis performed for the ERAS group according to hospital stay, with an optimal cut-off point of 36.99 HU.

In order to continue with the analysis, it was necessary to identify a single cut-off point for the muscle HU variable and use the same value in all scenarios to allow an objective comparison. To this end, every integer value from 34 to 41 HU was evaluated as a potential threshold. For each cut-off point, we assessed its ability to identify patients with poorer outcomes—both in terms of hospital stay and postoperative complications—in the ERAS and non-ERAS groups. Sensitivity and specificity values were calculated for each threshold, considering both prognostic outcomes in both groups. Mean values were also included to support global threshold selection.

The results presented in Table 5 indicate that lowering the HU threshold for muscle increases sensitivity for detecting patients at higher risk—both in terms of hospital stay and postoperative complications—while decreasing specificity. Based on the average values shown, 37 HU was selected as the optimal cut-off point, offering a trade-off between sensitivity (61.57%) and specificity (66.33%). Although this balance is acceptable, these values reflect only a moderate screening performance, though noticeably better than that of the MUST score.

To explore the potential impact of this threshold-based screening method on clinical outcomes, patients were classified into two subgroups (HU ≤ 37 vs. HU > 37) in both the Non-ERAS and ERAS cohorts. The previously described clinical, nutritional, and surgical variables were then compared between these subgroups. The corresponding findings are presented in Table 6. In the non-ERAS group, a screening method based on muscle radiodensity (HU) would have enabled clearer distinctions in clinical outcomes. Patients with HU ≤ 37 had a significantly shorter surgery time (median: 236 min, IQR: 216.0–277.0) compared to those with HU > 37 (median: 262.5 min, IQR: 219.5–316.25; *p* = 0.037), as well as fewer postoperative complications overall (15% vs. 33%; *p* = 0.036). They also experienced significantly fewer severe postoperative complications (Clavien–Dindo ≥ 3) (2% vs. 14%; *p* = 0.016) and showed a trend toward reduced nasogastric tube aspiration (10% vs. 23%; *p* = 0.09). In contrast, no significant differences emerged when using the MUST score, the SARC-F, or the GLIM criteria, suggesting limited screening capacity for these methods in this context. A significant difference in ECOG performance status was observed (*p* = 0.017), indicating a higher ECOG value for patients with HU ≤ 37. Patients with HU > 37 tended to be younger (median: 68.2 years, IQR: 61.0–76.68) compared to those with HU ≤ 37 (median: 76.3 years, IQR: 66.6–82.6; *p* = 0.001). In particular, there were no significant differences in cancer stage, metastasis status, or sex distribution, indicating that the two subpopulations were genuinely comparable.

It is important to note that no statistically significant differences in hospital length of stay were observed between the two groups defined by muscle radiodensity (median: 4 days, IQR: 3–12 for patients with HU ≤ 37 vs. median: 4 days, IQR: 3–5 for those with HU > 37; *p* = 0.73). This finding can be attributed to several factors. First, while the minimum hospital stay in both groups is 3 days, the maximum varies considerably, leading to a non-normal and skewed distribution. Although both groups share the same median (4 days), indicating that 50% of the patients were discharged within this time frame, a substantial difference is observed in the third quartile (Q3): 12 days in the HU ≤ 37 group versus 5 days in the HU > 37 group. This implies that 25% of the patients with lower muscle radiodensity (HU ≤ 37) remained hospitalized for more than 12 days. Although this difference is not statistically significant when comparing the overall distributions, it may still reflect a clinically relevant impact.

In the ERAS group, the comparison was less straightforward. Although there were no statistically significant differences in cancer stage or metastatic status, a significant difference in sex distribution suggests that these subgroups may not be fully comparable. Furthermore, between the time when the CT image was obtained (to calculate muscle HU values) and the date of surgery, the patients underwent the ERAS intervention, potentially influencing outcomes that would otherwise have remained unchanged. Despite these considerations, patients with HU ≤ 37 exhibited a significantly longer hospital stay (median: 4.0 days, IQR: 3.0–7.5 vs. median: 8 days, IQR: 4.5–13.5; *p* = 0.008) and a higher overall rate of postoperative complications (35% vs. 62%; *p* = 0.043). Although these differences did not reach statistical significance, there was a notable trend toward longer surgery duration (median: 253 min, IQR: 217.5–295.5 vs. median: 282.0 min, IQR: 248.75–330.0; *p* = 0.089) and increased nasogastric tube aspiration (18% vs. 41%; *p* = 0.064) in the HU ≤ 37 subgroup. No significant differences were found between the subgroups in MUST, SARC-F, or ECOG. However, the GLIM criteria identified a significantly higher prevalence of malnutrition in patients with HU > 37 compared to those with HU ≤ 37 (74% vs. 46%, *p* = 0.034).

Given the observed inconsistency between higher CT-based muscle quality (HU > 37) and increased rates of malnutrition as defined by the GLIM criteria, we conducted a post hoc analysis to explore the possible sources of this discrepancy. In our cohort, the GLIM phenotypic criterion was defined by the presence of at least one of the following: unintentional weight loss, low BMI, or low fat-free mass index (FFMI) as measured by bioimpedance.

We first evaluated the association between each individual GLIM component and CT-derived muscle radiodensity (dichotomized as >37 vs. ≤HU), using chi-squared tests. As shown in Table 7, no statistically significant association was found between muscle HU classification and weight loss, low BMI, or low FFMI considered separately.

Additionally, we compared continuous CT-derived variables (muscle HU, muscle area, and SMI) between patients classified as malnourished or not by each GLIM component. These results are presented in Table 8.

This analysis revealed that the GLIM classification, particularly when driven by BMI and fat-free mass index (FFMI), tends to align more closely with CT-derived measures of muscle quantity—namely muscle cross-sectional area and skeletal muscle index (SMI)—than with radiodensity.

Specifically, patients classified as malnourished according to the global GLIM criteria showed significantly lower muscle area (106.0 vs. 115.6 cm^2^, *p* = 0.023) and a trend toward lower SMI (*p* = 0.058), while no significant difference was observed in muscle HU (*p* = 0.072). This pattern was consistently observed across the weight loss and FFMI criteria, where differences in radiodensity were not significant, but muscle area and SMI were lower among malnourished patients. In contrast, the BMI-based criterion revealed a paradoxical finding: patients categorized as malnourished by low BMI exhibited higher muscle HU values (46.4 vs. 39.0, *p* = 0.012) despite having lower muscle area and SMI. This likely reflects the limitations of BMI in oncology populations, where weight loss can be driven by fat mass reduction without concurrent deterioration of muscle quality.

Collectively, these findings support the notion that GLIM criteria and CT-derived muscle radiodensity capture distinct and complementary dimensions of nutritional and functional status. GLIM emphasizes quantity-based parameters, while muscle HU reflects tissue quality, which is associated with myosteatosis and has independent prognostic implications. This underscores the value of incorporating opportunistic CT imaging into nutritional assessment frameworks, particularly in oncologic settings where traditional markers may not adequately reflect muscle integrity.

## 4. Discussion

This study evaluates the feasibility and clinical utility of using routine CT scans to screen for poor muscle quality in patients undergoing colorectal cancer surgery, either within or outside a prehabilitation program. To our knowledge, it is the first to demonstrate that muscle radiodensity (HU) derived from CT imaging outperforms the MUST questionnaire in identifying patients at higher risk of adverse postoperative outcomes. This approach may improve patient selection for Enhanced Recovery After Surgery (ERAS) strategies.

Our findings highlight the potential of Computed Tomography (CT) not only as a diagnostic tool but also as a powerful opportunistic method for nutritional screening in oncologic patients. Unlike traditional screening tools such as the Malnutrition Universal Screening Tool (MUST), CT-based evaluation allows for objective, quantitative assessment of muscle quality through radiodensity values expressed in Hounsfield Units (HUs) [34,35]. This approach offers several advantages: it leverages imaging that is already part of routine oncologic workups, it does not require additional procedures or patient burden, and it provides clinically relevant information without increasing costs [36,37].

One of the most relevant findings of this study is the limited concordance between CT-assessed muscle quality and the nutritional classification based on MUST or GLIM criteria. While 40% of our cohort presented with low muscle quality (HU ≤ 37), only a small proportion of these patients were identified as malnourished by the MUST tool.

This discrepancy highlights an inherent limitation of MUST: the lack of direct body composition parameters, which may lead to false negatives in patients with hidden sarcopenia, especially in oncologic settings where BMI does not always reflect nutritional status adequately [7,38].

Additionally, although muscle mass is a core phenotypic criterion in the GLIM framework, the proportion of patients diagnosed with malnutrition according to GLIM was similar across both muscle quality subgroups (60% with HU > 37 vs. 56% with HU ≤ 37; *p* = 0.755). This unexpected result may be explained by the fact that the GLIM criteria focus solely on muscle quantity, using indirect methods with limited accuracy. In contrast, CT-derived muscle radiodensity reflects muscle quality, which appears to be a more reliable indicator of the patient’s nutritional reserve.

These findings prompt reflection on the role of muscle quality in nutritional assessment, particularly in cancer surgery [39]. While the GLIM criteria represent a major step forward by including muscle mass as a diagnostic factor, our study shows that patients with high muscle radiodensity (HU > 37) may still be classified as malnourished. This apparent mismatch may stem from the inability of indirect methods to distinguish functional muscle from fat-infiltrated tissue. In this context, CT-based evaluation of muscle quality offers complementary and clinically valuable information, as it more precisely reflects the metabolic and functional muscle reserve. In selected populations, such as cancer patients, incorporating muscle quality metrics may improve diagnostic accuracy and enable more tailored risk stratification [40,41].

Stratification by the HU threshold also revealed clinically relevant trajectories. In the non-ERAS group, patients with low muscle quality (HU ≤ 37) showed higher rates of postoperative complications (32% vs. 15%, *p* = 0.036), more complications per patient (2.3 vs. 1.2, *p* = 0.04), and longer hospital stays (8 vs. 4 days, *p* = 0.03). These associations were independent of sex, age, and greater need for nutritional intervention. Similar trends were found in the ERAS subgroup, although attenuated, likely due to prehabilitation effects occurring between the CT assessment and the surgery.

These data further support the clinical utility of muscle radiodensity as a discriminative and feasible tool to guide patient selection for prehabilitation. Once automated, this measurement could be seamlessly integrated into clinical workflows. In summary, CT-based muscle quality assessment outperforms traditional screening tools such as MUST in identifying patients at nutritional risk and likely to benefit from tailored perioperative care.

Nevertheless, while the 37 HU threshold demonstrated promising discriminatory performance within our cohort, its external validity remains to be confirmed. To this end, we have initiated similar analyses in other oncological populations, and intend to further explore the applicability and clinical relevance of this radiodensity-based marker in broader surgical settings. Additionally, given the exploratory nature of this study and the aim to evaluate the individual discriminative capacity of specific CT-derived metrics, multivariate analysis was not performed. A Variance Inflation Factor (VIF) analysis was conducted to confirm the absence of multicollinearity among the selected variables. Future studies seeking to build predictive models or risk stratification tools should incorporate multivariate approaches to adjust for potential confounders and improve generalizability.

## 5. Conclusions

CT-derived muscle radiodensity offers superior predictive value over traditional screening tools such as MUST to identify colorectal cancer patients at risk of poor postoperative outcomes. Its opportunistic use leverages routine imaging without additional burden, providing objective, reproducible data on muscle quality, a key determinant of surgical prognosis. Implementing this parameter in prehabilitation selection can improve patient stratification, optimize resource allocation, and improve clinical outcomes. Further validation in larger prospective cohorts is warranted.

## Figures and Tables

**Figure 1 nutrients-17-02264-f001:**
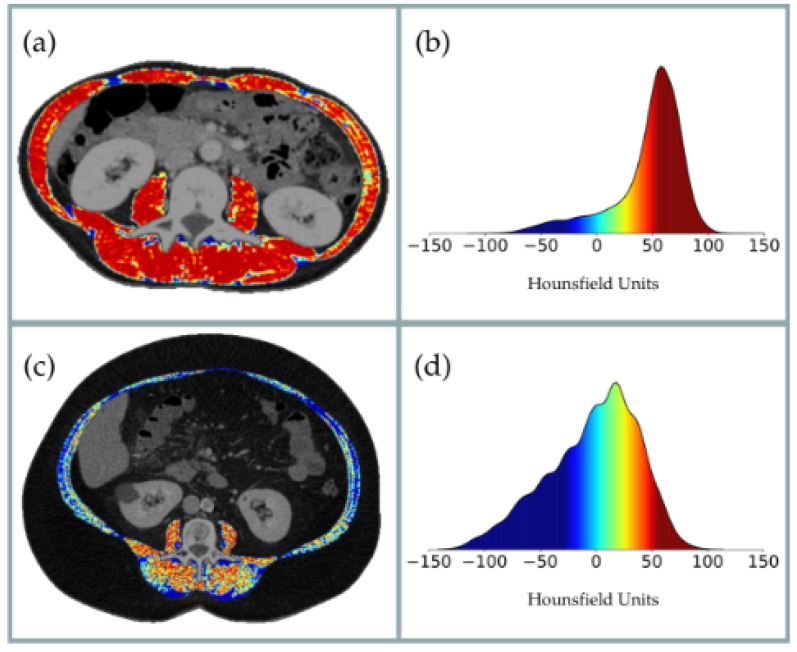
Radiodensity distribution of skeletal muscle tissue segmented from abdominal CT images in two representative patients. Panels (**a**,**c**) show axial CT slices with segmented skeletal muscle tissue, where each pixel is color-coded based on its radiodensity in Hounsfield Units (HUs). In this color scale, warm colors (yellow to red) represent higher muscle density and better muscle quality, while cool colors (blue to dark blue) represent lower muscle density and greater fat infiltration. In panel (**a**), a predominance of warm colors indicates high muscle quality, reflecting well-preserved muscle with minimal fat infiltration. In contrast, panel (**c**) displays a predominance of cool tones, indicating poor muscle quality with significant fatty infiltration and replacement. Panels (**b**,**d**) present the corresponding histograms of HU distribution for each patient, using the same color scale as in panels (**a**,**c**), providing a quantitative representation of muscle radiodensity.

**Figure 2 nutrients-17-02264-f002:**
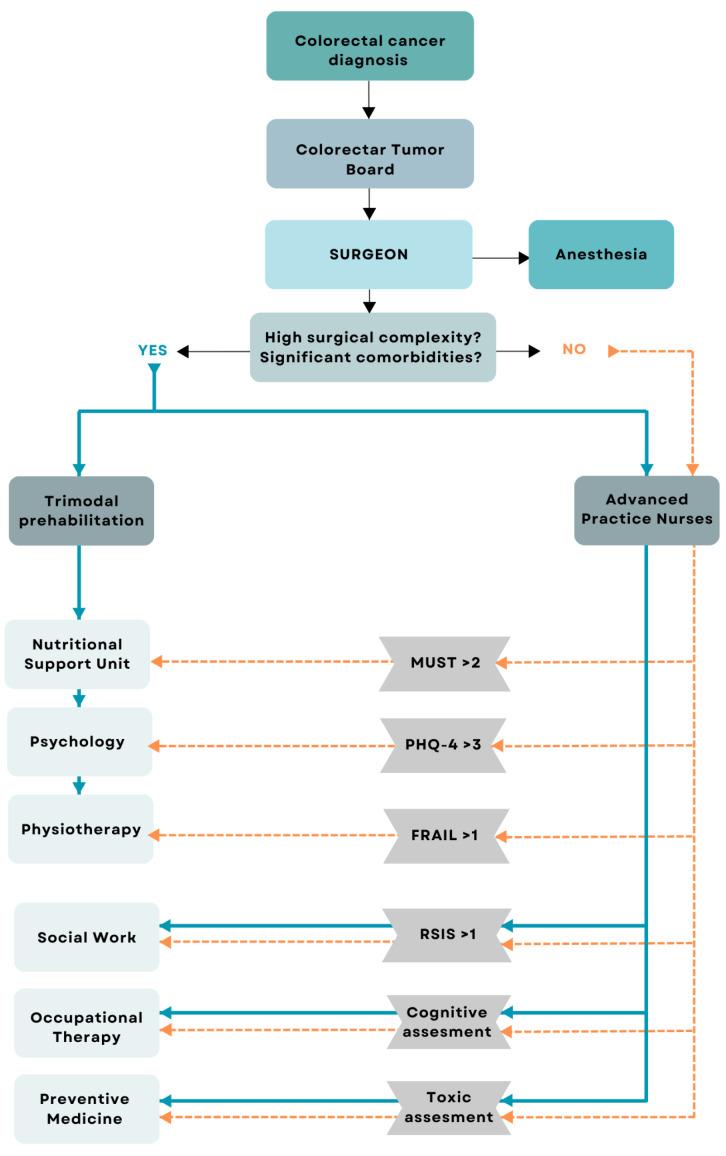
Flowchart outlining the referral and screening process for inclusion in the ERAS prehabilitation program. MUST—Malnutrition Universal Screening Tool; FRAIL—Fatigue, Resistance, Ambulation, Illnesses, and Loss of Weight; PHQ-4—Patient Health Questionnaire-4; ERAS—Enhanced Recovery After Surgery; RSIS—Rapid Social Intervention Screening tool.

**Table 1 nutrients-17-02264-t001:** General characteristics of the sample.

	Whole Sample n=201	ERAS Group n=71 (35.3%)	Non-ERAS Group n=130 (64.7%)	*p* Value
**Age (years)**	73.0 (64.8, 81.6)	78.7 (69.4, 86.4)	70.3 (63.5, 77.9)	**<0.001**
**Sex—Men**	120 (60%)	41 (58%)	79 (61%)	0.789
**Sex—Women**	81 (40%)	30 (42%)	51 (39%)	0.789
**BMI (kg/m^2^)—Men**	26.7 (24.2, 29.4)	25.7 (23.8, 27.8)	27.4 (24.2, 29.7)	0.114
**BMI (kg/m^2^)—Women**	25.9 (23.5, 29.1)	24.9 (22.7, 26.7)	26.8 (24.2, 30.3)	0.052
**Cancer stage 1–2**	104 (52%)	35 (49%)	69 (53%)	0.632
**Cancer stage 3–4**	84 (42%)	32 (45%)	52 (40%)	
**Metastases (yes)**	9.0 (4%)	4.0 (6%)	5.0 (4%)	0.723
**Time of surgery**	250.5 (218.5, 290.0)	262 (228.0, 307.0)	240 (217.5, 286.5)	**0.041**
**Hospital stay (days)**	4 (3.0, 8.0)	6 (4.0, 11.0)	4.0 (3.0, 5.25)	**<0.001**
**Post-op complications**	62 (31%)	35 (49%)	27 (21%)	**<0.001**
**SNG aspiration**	40 (20%)	21 (30%)	19 (15%)	**0.019**
**Clavien Dindo classification** ≥3	17 (8%)	9 (13%)	8 (6%)	0.186
**Exitus**	16 (8%)	10 (14%)	6 (5%)	**0.036**
**Risk of malnutrition (MUST screening)**	56 (28%)	27 (38%)	29 (22%)	**0.027**
**Risk of sarcopenia (SARC-F screening)**	28 (14%)	14 (20%)	14 (11%)	0.124
**GLIM**	117 (58%)	42 (59%)	75 (58%)	0.959
**ECOG**				**0.001**
**0**	146 (73%)	41 (58%)	105 (81%)	**0.0009**
**1**	40 (20%)	19 (27%)	21 (16%)	0.106
**2**	13 (6%)	9 (13%)	4 (3%)	**0.014**
**3**	2 (1%)	2 (3%)	0 (0%)	0.124

**Abbreviations:** ERAS = Enhanced Recovery After Surgery; BMI = Body Mass Index; SNG = nasogastric tube; MUST = Malnutrition Universal Screening Tool; SARC-F = Strength, Assistance in walking, Rise from a chair, Climb stairs and Falls; GLIM = Global Leadership Initiative on Malnutrition; ECOG = Eastern Cooperative Oncology Group. **Bold values indicate a significant *p*-value < 0.05.**

**Table 2 nutrients-17-02264-t002:** Predictive performance of the MUST Score (MUST = 0 vs. MUST > 0) in relation to hospital stay and postoperative complications across ERAS and Non-ERAS groups.

MUST Analysis. MUST = 0 Versus MUST > 0 Based on Postoperative Complications				
**Group** & **MUST Score**	**No Complications**	**Complications**	**Sensitivity**	**Specificity**
**ERAS = 0**	28	27	22.86	77.78
**ERAS > 0**	8	8		
**Non-ERAS = 0**	93	23	14.81	90.29
**Non-ERAS > 0**	10	4		
**MUST Analysis. MUST = 0 Versus MUST > 0 Based on Hospital Stay**				
**Group** & **MUST Score**	**Normal Stay**	**Long Stay**	**Sensitivity**	**Specificity**
**ERAS = 0**	28	19	20.83	75.68
**ERAS > 0**	9	5		
**Non-ERAS = 0**	96	14	26.32	92.31
**Non-ERAS > 0**	8	5		

**Abbreviations:** MUST = Malnutrition Universal Screening Tool; ERAS = Enhanced Recovery After Surgery.

**Table 3 nutrients-17-02264-t003:** ROC curve analysis results for ERAS and non-ERAS groups based on postoperatory complications.

Variable	ERAS—Complications		Non-ERAS—Complications	
	**AUC/Optimal Threshold**	**Sens/Spec**	**AUC/Optimal Threshold**	**Sens/Spec**
**Muscle (HU)**	0.692/34.97	60/75	0.659/39.71	67/64
**Height**	0.586/1.65	66/58	0.535/1.63	52/64
**SMI**	0.556/36.14	63/64	0.640/39.86	74/59
**BMI**	0.548/24.69	49/69	0.556/27.43	67/56
**Age**	0.539/78.74	57/56	0.501/70.8	59/50
**Muscle (cm^2^)**	0.519/96.15	57/58	0.635/106.01	63/61
**Weight**	0.508/71	49/56	0.501/73	59/55

**Abbreviations:** ERAS = Enhanced Recovery After Surgery; AUC = Area Under the Curve; HUs = Hounsfield Units; SMI = Skeletal Muscle Index; BMI = Body Mass Index; Sens = Sensitivity; Spec = Specificity.

**Table 4 nutrients-17-02264-t004:** ROC curve analysis results for ERAS and non-ERAS groups based on hospital stay.

Variable	ERAS—Hospital Stay		Non-ERAS—Hospital Stay	
	**AUC/Optimal Threshold**	**Sens/Spec**	**AUC/Optimal Threshold**	**Sens/Spec**
**Muscle (HU)**	0.688/36.99	71/62	0.620/34.46	58/79
**Height**	0.606/1.65	67/57	0.570/1.70	53/64
**SMI**	0.600/36.14	71/62	0.621/39.86	68/57
**BMI**	0.600/24.69	54/73	0.529/25.48	47/65
**Age**	0.586/76.97	75/51	0.569/64.69	47/72
**Muscle (cm^2^)**	0.533/96.15	58/59	0.584/104.49	58/63
**Weight**	0.524/68	54/54	0.506/72	53/58

**Abbreviations:** ERAS = Enhanced Recovery After Surgery; AUC = Area Under the Curve; HUs = Hounsfield Units; SMI = Skeletal Muscle Index; BMI = Body Mass Index; Sens = Sensitivity; Spec = Specificity.

**Table 5 nutrients-17-02264-t005:** Diagnostic performance of muscle density thresholds (Hounsfield units) in predicting hospital stay and postoperative complications in ERAS and non-ERAS groups.

	ERAS				Non-ERAS					
**Muscle (HU)**	**Hospital Stay**		**Complications**		**Hospital Stay**		**Complications**		**Sensitivity**	**Specificity**
**Threshold**	**Sens**	**Spec**	**Sens**	**Spec**	**Sens**	**Spec**	**Sens**	**Spec**	**Mean (%)**	**Mean (%)**
34	50	78.38	48.57	80.56	52.63	79.81	44.44	80.58	48.91	79.83
35	58.33	70.27	60	75	57.89	76.92	48.15	77.67	56.09	74.97
36	62.5	67.57	60	69.44	57.89	72.12	51.85	73.79	58.06	70.73
37	70.83	62.16	65.71	61.11	57.89	70.19	51.85	71.84	61.57	66.33
38	70.83	56.76	65.71	55.56	57.89	67.31	51.85	68.93	61.57	62.14
39	70.83	54.05	68.57	52.78	57.89	63.46	55.56	66.02	63.21	59.08
40	70.83	48.65	68.57	47.22	57.89	57.69	66.67	62.14	65.99	53.93
41	75	40.54	74.29	41.67	57.89	48.08	70.37	53.4	69.39	45.92

**Abbreviations:** ERAS = Enhanced Recovery After Surgery; HU = Hounsfield Units; Sens = Sensitivity; Spec = Specificity.

**Table 6 nutrients-17-02264-t006:** General characteristics of ERAS and non-ERAS groups using muscle HU 37 as cut-off value.

	ERAS Group				Non-ERAS Group			
	**Whole Sample**	**Muscle HU > 37**	**Muscle HU ≤37**	* **p** *	**Whole sample**	**Muscle HU > 37**	**Muscle HU ≤37**	* **p** *
	***n*** **= 71**	***n*** **= 34 (47.9%)**	***n** * **= 37 (52.1%)**	**Value**	***n*** **= 130**	***n*** **= 87 (66.9%)**	***n*** **= 43 (33.1%)**	**Value**
Age (years)	78.7 (69.4, 86.4)	76.0 (67.3, 85.4)	81.5 (72.9, 86.9)	0.106	70.3 (63.5, 77.9)	68.2 (61.0, 76.8)	76.3 (66.6, 82.6)	**0.001**
Sex—Men	41 (58%)	15 (44%)	26 (70%)	**0.047**	79 (61%)	51 (59%)	28 (65%)	0.601
Sex—Women	30 (42%)	19 (56%)	11 (30%)	**0.047**	51 (39%)	36 (41%)	15 (35%)	0.601
BMI (kg/m^2^)—Men	25.7 (23.8, 27.8)	25.0 (24.1, 27.3)	26.2 (23.4, 27.9)	0.818	27.4 (24.2, 29.7)	25.6 (23.7, 29.3)	28.8 (27.3, 30.4)	**0.008**
BMI (kg/m^2^)—Women	24.9 (22.7, 26.7)	24.1 (22.2, 25.6)	25.8 (24.4, 30.5)	**0.043**	26.8 (24.2, 30.3)	26.4 (23.6, 29.1)	26.8 (25.0, 33.3)	0.189
Cancer stage				0.18				0.904
Cancer stage 1–2	35 (49%)	14 (41%)	21 (57%)		69 (53%)	47 (54%)	22 (51%)	
Cancer stage 3–4	32 (45%)	19 (56%)	13 (35%)		52 (40%)	34 (39%)	18 (42%)	
Metastases (yes)	4.0 (6%)	1.0 (3%)	3.0 (8%)	0.615	5.0 (4%)	2.0 (2%)	3.0 (7%)	0.331
Time of surgery	262 (228.0, 307.0)	253 (217.5, 295.5)	282.0 (248.75, 330.0)	0.089	240 (217.5, 286.5)	236 (216.0, 277.0)	262.5 (219.5, 316.25)	**0.037**
Hospital stay (days)	6 (4.0, 11.0)	4.0 (3.0, 7.5)	8 (4.5, 13.5)	**0.008**	4.0 (3.0, 5.25)	4 (3.0, 5.0)	4 (3.0, 12.0)	0.73
Post-op complications	35 (49%)	12 (35%)	23 (62%)	**0.043**	27 (21%)	13 (15%)	14 (33%)	**0.036**
SNG aspiration	21 (30%)	6 (18%)	15 (41%)	0.064	19 (15%)	9 (10%)	10 (23%)	0.09
Clavien Dindo classification ≥3	9 (13%)	3 (9%)	6 (16%)	0.482	8 (6%)	2 (2%)	6 (14%)	**0.016**
Exitus	10 (14%)	5 (15%)	5 (14%)	1	6 (5%)	4 (5%)	2 (5%)	1
Malnourished (MUST screening)	27 (38%)	14 (41%)	13 (35%)	0.78	29 (22%)	17 (20%)	12 (28%)	0.393
Risk of sarcopenia (SARC-F screening)	14 (20%)	6 (18%)	8 (22%)	0.903	14 (11%)	7 (8%)	7 (16%)	0.261
GLIM	42 (59%)	25 (74%)	17 (46%)	**0.034**	75 (58%)	47 (54%)	28 (65%)	0.31
ECOG				0.145				**0.017**
0	41 (58%)	24 (71%)	17 (46%)	0.063	105 (81%)	76 (87%)	29 (67%)	**0.013**
1	19 (27%)	7 (21%)	12 (32%)	0.391	21 (16%)	10 (11%)	11 (26%)	0.072
2	9 (13%)	3 (9%)	6 (16%)	0.482	4 (3%)	1 (1%)	3 (7%)	0.105
3	2 (3%)	0 (0%)	2 (5%)	0.494	0	0	0	

**Abbreviations:** ERAS = Enhanced Recovery After Surgery; HUs = Hounsfield Units; BMI = Body Mass Index; SNG = nasogastric tube; MUST = Malnutrition Universal Screening Tool; SARC-F = Strength, Assistance in walking, Rise from a chair, Climb stairs and Falls; GLIM = Global Leadership Initiative on Malnutrition; ECOG = Eastern Cooperative Oncology Group. **Bold values indicate a significant *p*-value < 0.05.**

**Table 7 nutrients-17-02264-t007:** Association between GLIM criteria components and muscle radiodensity threshold (HU > 37 vs. ≤37).

GLIM Criteria	Total (*n* = 201)	Muscle HU > 37 (*n* = 121)	Muscle HU ≤ 37 (*n* = 80)	*p*-Value
At least one criterion met	117 (58%)	72 (60%)	45 (56%)	0.755
Weight loss criterion	104 (52%)	60 (50%)	44 (55%)	0.543
BMI-based criterion	12 (6%)	10 (8%)	2 (2%)	0.129
FFMI-based criterion	27 (13%)	20 (17%)	7 (9%)	0.170

**Abbreviations:** GLIM = Global Leadership Initiative on Malnutrition; FFMI = Fat-Free Mass Index; HUs = Hounsfield Units.

**Table 8 nutrients-17-02264-t008:** Comparison of CT-derived body composition variables by GLIM criteria components.

GLIM Criteria	CT Variable	Total	Not Malnourished	Malnourished	*p*-Value
At least one criterion met		(*n* = 201)	(*n* = 84)	(*n* = 117)	
	Muscle HU	39.42 ± 9.94	37.93 ± 9.20	40.49 ± 10.34	0.072
	Muscle Area (cm^2^)	110.03 ± 29.66	115.61 ± 30.15	106.02 ± 28.76	0.023
	SMI (cm^2^/m^2^)	39.92 ± 8.61	41.28 ± 8.98	38.95 ± 8.23	0.058
Weight loss criterion		(*n* = 201)	(*n* = 97)	(*n* = 104)	
	Muscle HU	39.42 ± 9.94	39.12 ± 9.38	39.70 ± 10.47	0.677
	Muscle Area (cm^2^)	110.03 ± 29.66	113.20 ± 30.38	107.07 ± 28.80	0.144
	SMI (cm^2^/m^2^)	39.92 ± 8.61	40.53 ± 8.84	39.35 ± 8.39	0.335
BMI-based criterion		(*n* = 201)	(*n* = 189)	(*n* = 12)	
	Muscle HU	39.42 ± 9.94	38.98 ± 9.93	46.41 ± 7.37	0.012
	Muscle Area (cm^2^)	110.03 ± 29.66	111.22 ± 29.58	91.31 ± 25.10	0.024
	SMI (cm^2^/m^2^)	39.92 ± 8.61	40.41 ± 8.55	32.27 ± 5.54	0.001
FFMI-based criterion		(*n* = 201)	(*n* = 134)	(*n* = 27)	
	Muscle HU	39.42 ± 9.94	40.29 ± 10.28	41.48 ± 6.95	0.566
	Muscle Area (cm^2^)	110.03 ± 29.66	114.69 ± 29.46	98.94 ± 24.62	0.010
	SMI (cm^2^/m^2^)	39.92 ± 8.61	41.22 ± 8.58	36.54 ± 6.31	0.008

**Abbreviations:** GLIM = Global Leadership Initiative on Malnutrition; FFMI = Fat-Free Mass Index; HU = Hounsfield Units; SMI = Skeletal Muscle Index.

## Data Availability

The original contributions presented in the study are included in the article, further inquiries can be directed to the corresponding authors.

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
