# Peer review of "Prognostic Value of Computed Tomography-Derived Muscle Density for Postoperative Complications in Enhanced Recovery After Surgery (ERAS) and Non-ERAS Patients"

_nutrients, 2025, doi:10.3390/nu17142264_

Round 1
Reviewer 1 Report
Comments and Suggestions for Authors
Dear Authors,
Patient selection for prehabilitation is a key topic in surgical oncology. The use of routine CT is not only ingenious and practical, but, as you show in your article, CT-HU (37 HU threshold) outperforms MUST in terms of sensitivity and specificity (AUC 0.62-0.69 vs. MUST sensitivity <27%) and identifies groups with significantly different postoperative trajectories.
My comments on your article are listed below:
- Please delete the funding information in the abstract. This statement is already present at the bottom of your article.
- It is advisable to perform a validation analysis of the 37 HU threshold by either dividing the cohort into training and testing datasets or utilizing an independent patient sample. If this is not possible, discuss the limitations of non-validation in the discussion section.
- Multivariate analysis would have controlled for sex, age, and other variables that differed between groups. Please explain why you did not use this type of analysis.
- More thorough discussion is required in light of the surprising finding that individuals with higher muscle quality, as determined by CT, were more frequently categorized as malnourished by GLIM. Is this an anomaly of the oncology cohort or a misclassification by BIA?
- Despite the list of citations at the end of the article, I do not see references in the discussion text. Please supplement the citations and deepen the discussion, which, as it stands, is more like a presentation of results. You might want to add a short paragraph on the possible implications of the results for the daily practice of surgeons
Best regards,
The reviewer
Author Response
Response to Reviewers
We would like to express our sincere gratitude to the reviewers for their thoughtful and constructive comments on our manuscript. We deeply appreciate the time and effort dedicated to the review process, as well as the valuable suggestions provided. These observations have significantly contributed to improving the clarity, methodological rigor, and overall quality of our work. Below, we provide a point-by-point response to each of the reviewers’ comments. All changes made in the manuscript in response to the reviewers’ suggestions are highlighted in blue for ease of reference.
Revisor 1
Dear Authors,
Patient selection for prehabilitation is a key topic in surgical oncology. The use of routine CT is not only ingenious and practical, but, as you show in your article, CT-HU (37 HU threshold) outperforms MUST in terms of sensitivity and specificity (AUC 0.62-0.69 vs. MUST sensitivity <27%) and identifies groups with significantly different postoperative trajectories.
My comments on your article are listed below:
- Please delete the funding information in the abstract. This statement is already present at the bottom of your article.
We thank the reviewer for this observation. As suggested, the funding information has been removed from the abstract, since it is already included in the appropriate section at the end of the manuscript. - It is advisable to perform a validation analysis of the 37 HU threshold by either dividing the cohort into training and testing datasets or utilizing an independent patient sample. If this is not possible, discuss the limitations of non-validation in the discussion section.
We appreciate the reviewer’s suggestion regarding validation of the 37 HU threshold. However, we would like to clarify that no predictive model was trained or fitted in our study. The aim was to assess the discriminatory performance of individual body composition variables—particularly CT-derived muscle radiodensity—in relation to postoperative outcomes, based on predefined clinical endpoints. The 37 HU threshold was derived using ROC curve analysis within this cohort, as a means of identifying a clinically meaningful cut-off point, rather than for predictive modeling purposes.
Given the observational nature of the study and absence of any model development, splitting the sample into training and test sets was not applicable. That said, we fully agree on the importance of external validation of this threshold in other patient populations. We have already initiated a similar study in ovarian cancer patients, and we plan to expand this line of research to confirm the generalizability of our findings across different oncological settings.
To address this point, we have added a statement at the end of the discussion acknowledging this limitation and outlining our future validation efforts.
- Multivariate analysis would have controlled for sex, age, and other variables that differed between groups. Please explain why you did not use this type of analysis.
We thank the reviewer for this important observation. In our study, the primary objective was not to construct a predictive model, but rather to evaluate the individual discriminatory power of specific body composition variables—particularly CT-derived muscle radiodensity—for identifying patients at risk of poor postoperative outcomes.
To ensure the independence of the variables analyzed, we performed a Variance Inflation Factor (VIF) analysis including age, weight, height, muscle area (cm²), and muscle radiodensity (HU). All variables exhibited VIF values well below the conventional threshold of 3, confirming the absence of multicollinearity and supporting the inclusion of these parameters as independent variables in our exploratory analysis.
Given the observational and exploratory nature of the study, we chose to focus on univariate analyses to assess the discriminative performance of each variable. While we acknowledge that multivariate models can provide additional insights, we believe this would represent a distinct objective—namely, the development of a predictive tool—which was beyond the scope of this initial investigation.
Nevertheless, we agree that future studies aimed at building risk prediction models should incorporate multivariate approaches to adjust for potential confounders and to enhance generalizability. To address this point, a clarification has been added at the end of the Discussion section of the manuscript.
- More thorough discussion is required in light of the surprising finding that individuals with higher muscle quality, as determined by CT, were more frequently categorized as malnourished by GLIM. Is this an anomaly of the oncology cohort or a misclassification by BIA?
We thank the reviewer for raising this important point. To explore this discrepancy, we performed an additional analysis in which we examined each GLIM component separately and assessed their relationship with CT-derived muscle radiodensity. The results of this analysis have been included at the end of the Results section.
These findings support the notion that GLIM criteria and CT-derived muscle radiodensity assess distinct aspects of nutritional and body composition status. While GLIM primarily reflects the quantity of body mass—particularly through variables such as BMI, weight loss, and fat-free mass index—muscle HU quantifies the quality of muscle tissue, specifically its radiological density. This parameter is influenced by intramuscular fat infiltration and is closely linked to muscle metabolic function. The analysis revealed moderate agreement between GLIM and CT-derived muscle area and SMI, but no clear relationship with muscle HU. This suggests that muscle quality, as measured by HU, provides complementary and independent information not captured by conventional malnutrition criteria, highlighting the added value of opportunistic CT analysis in nutritional assessment. - Despite the list of citations at the end of the article, I do not see references in the discussion text. Please supplement the citations and deepen the discussion, which, as it stands, is more like a presentation of results. You might want to add a short paragraph on the possible implications of the results for the daily practice of surgeons.
We thank the reviewer for this suggestion. Additional references have been included and the discussion has been expanded accordingly to provide greater context and highlight the clinical relevance of the findings.
Best regards,

Reviewer 2 Report
Comments and Suggestions for Authors
Evaluation of manuscript nutrients-3720806
This is an interesting manuscript about muscle mass assessment using CT in rehabilitation. Below, I present my considerations regarding the study to assist the authors.
Initially, I would like to advise the authors to be cautious with acronyms. It is not ideal to use them in the title; I recommend replacing "ERAS" with "Enhanced Recovery After Surgery." Additionally, when introducing an acronym for the first time, it should be explained. In the Abstract, "SARC-F" and "GLIM" were not defined.
The introduction is well-written; however, there is a lack of connection between the paragraphs. The second paragraph discusses cancer, specifically colorectal cancer, but no clear link is established between the first and second paragraphs. It is only in the Methods section that the authors mention the sample consists of colorectal cancer patients. This could be resolved if the authors revise the objective to also include the characteristics of the patients being assessed. I also recommend adding the hypothesis at the end of the introduction.
I suggest that the authors present the ethical criteria at the beginning of the Methods section.
Tables and figures should be self-explanatory, so all acronyms must be defined in the notes.
The results are well-presented. However, in the Discussion, I advise the authors to avoid repeating the results. Please review this section. The Discussion should further explain the study's findings based on previous research. I recommend rewriting it to demonstrate how the current results can enhance clinical practice for patients undergoing colorectal cancer surgery.
Author Response
We would like to express our sincere gratitude to the reviewers for their thoughtful and constructive comments on our manuscript. We deeply appreciate the time and effort dedicated to the review process, as well as the valuable suggestions provided. These observations have significantly contributed to improving the clarity, methodological rigor, and overall quality of our work. Below, we provide a point-by-point response to each of the reviewers’ comments. All changes made in the manuscript in response to the reviewers’ suggestions are highlighted in blue for ease of reference.
The results are well-presented. However, in the Discussion, I advise the authors to avoid repeating the results. Please review this section. The Discussion should further explain the study's findings based on previous research. I recommend rewriting it to demonstrate how the current results can enhance clinical practice for patients undergoing colorectal cancer surgery.
We thank the reviewer for the detailed and constructive suggestions, which have helped us improve the clarity and structure of the manuscript. All comments have been carefully addressed and the corresponding changes have been made in the text. These modifications are highlighted in blue for ease of review.

Round 2
Reviewer 1 Report
Comments and Suggestions for Authors
Dear Authors,
Thank you very much for your satisfactory answers to my suggestions.
Best regards,
The reviewer.
Reviewer 2 Report
Comments and Suggestions for Authors
Of the points that were suggested in my review, all were met. Therefore, I consider the manuscript suitable for publication.